# Early-life environment shapes claw bilateral asymmetry in the European lobster (*Homarus gammarus*)

Lorenzo Latini[1,‡], Gioia Burini[2], Valeria Mazza[2], Giacomo Grignani[2], Riccardo De Donno[2], Eleonora Bello[2], Elena Tricarico[1], Stefano Malavasi[3], Giuseppe Nascetti[2], Daniele Canestrelli[2,*] and Claudio Carere[2,*]

## ABSTRACT

Developmental plasticity refers to an organism's ability to adjust its development in response to changing environmental conditions, leading to changes in behaviour, physiology, or morphology. This adaptability is crucial for survival and helps organisms to cope with environmental challenges throughout their lives. Understanding the mechanisms underlying developmental plasticity, particularly how environmental and ontogenetic factors shape functional traits, is fundamental for both evolutionary biology and conservation efforts. In this study we investigated the effects of early-life environmental conditions on the development of claw asymmetry in juvenile European lobsters (*Homarus gammarus*, N=244), a functional trait essential for survival and ecological success. Juveniles were randomly divided between four different rearing conditions characterized by the presence or absence of physical enrichments (e.g. substrate and shelters), which were introduced at different developmental stages in separated groups to assess the timing and nature of their effect. Results revealed that exposure to substrate alone, without additional stimuli, consistently promoted claw asymmetry, regardless of the timing of its introduction, while the 6th developmental stage emerged as the critical period for claw differentiation. By identifying the environmental factors that influence developmental outcomes in lobsters, and the timing of these effects, this study improves our understanding of developmental plasticity and offers valuable insights for optimizing conservation aquaculture and reintroduction strategies.

KEY WORDS: Bilateral asymmetry, Claw differentiation, Conservation aquaculture, Developmental plasticity, Phenotypic development, Ontogeny

## INTRODUCTION

Developmental plasticity, shaped by the dynamic interaction between genes and the environment, enables organisms to adjust their developmental pathways, leading to changes in traits like behaviour, physiology, and morphology (Raff and Wolpert, 1996; West-Eberhard, 2003). This ability has important ecological and evolutionary implications, helping individuals and populations to survive and persist under fluctuating environmental conditions (West-Eberhard, 2003; De Witt and Scheiner, 2004). Environmental factors, especially those experienced during critical developmental periods, can regulate gene expression, allowing a single genotype to produce different phenotypes (Holloway, 2002; Pigliucci, 2005). Such environmentally induced responses are mediated by epigenetic mechanisms (such as DNA methylation, histone modification, and the role of non-coding RNAs) that modify transcriptional activity without altering the underlying DNA sequence (Jablonka and Lamb, 2007; Berger et al., 2009). These mechanisms enable organisms to adapt to external stimuli by activating or silencing genes, leading to a variety of phenotypic variations (Gibney and Nolan, 2010; Moore et al., 2013). For example, stickleback (*Gasterosteus aculeatus*) individuals exposed to different salinity levels develop distinct gill and kidney structures to optimize ion exchange, enhancing their survival in both freshwater and marine habitats (Hasan et al., 2017). Similarly, larvae of European tree frogs (*Hyla arborea*) exposed to predator cues grow larger tail fins to evade predators, while those in predator-free environments prioritize rapid growth to boost reproductive potential (Luquet et al., 2011).

However, developmental plasticity has its limits, particularly when environmental conditions are too extreme or lack critical developmental cues (Chevin and Hoffmann, 2017). In such cases, plasticity may fail to fully compensate for stress, leading to maladaptive traits or impaired development (Schwab et al., 2019; Klingenberg, 2019). One well-documented indicator of developmental instability is fluctuating asymmetry, which refers to small, random deviations from perfect bilateral symmetry (Palmer and Strobeck, 1986; Zakharov et al., 2020). Elevated levels of fluctuating asymmetry are often observed in organisms subjected to environmental stressors, such as contamination or nutritional deficiency, reflecting an inability to maintain stable development (Møller and Swaddle, 1997; De Coster et al., 2013; Hammelman et al., 2020). For instance, mussels (*Mytilus galloprovincialis*) translocated in polluted environments or exposed to elevated aerial temperatures show higher levels of fluctuating asymmetry in shell shape compared to control groups (Nishizaki et al., 2015; Scalici et al., 2017). In the blue damselfish (*Chrysiptera cyanea*) individuals raised in structurally simple environments develop asymmetrical otoliths, which impair navigation abilities and reduce success in locating suitable settlement habitats (Gagliano et al., 2008). These examples highlight how environmental stressors or lack of critical stimuli interfere with the normal developmental processes that ensure symmetrical growth, leading to greater fluctuating asymmetry and compromised developmental stability.

[1]Department of Biology, University of Florence, via Madonna del Piano 6, 50019 Sesto Fiorentino, Italy. [2]Department of Ecological and Biological Sciences, University of Tuscia, Largo dell'Università, 01100, Viterbo, Italy. [3]Department of Environmental Sciences, Informatics and Statistics, Ca' Foscari University of Venice, Via Torino 155, 30172, Venice, Italy.
*These authors contributed equally to this work

‡Author for correspondence (lorenzo.latini.94@gmail.com)

L.L., 0000-0003-4698-2532; V.M., 0000-0002-1634-3417; G.G., 0000-0002-6457-405X; E.B., 0000-0001-8501-5858; E.T., 0000-0002-7392-0794; S.M., 0000-0001-8071-1010; G.N., 0000-0002-1159-8103; D.C., 0000-0001-9351-4972; C.C., 0000-0003-1644-2113

While in bilaterally symmetrical organisms, deviations from perfect symmetry indicate developmental instability/stress, in bilaterally asymmetrical organisms, the development of asymmetrical traits reflects optimal developmental conditions. Nevertheless, even in these organisms, early-life environmental factors can significantly influence the expression of genetically programmed asymmetry, potentially leading to less functional or even symmetrical traits (Palmer, 2016). A notable case is the clawed lobster (both *Homarus americanus* and *Homarus gammarus*), which typically develops pronounced claw asymmetry during its early benthic stages (Templeman, 1935; Govind and Pearce, 1989). During this critical developmental window, experience – specifically the consistent use of one claw over the other – triggers the formation of a larger, stronger 'crusher' claw, used to break hard prey, while the other claw, referred to as 'cutter', becomes smaller and more agile, functioning as a pincer for precise tasks (Govind, 1989; Govind and Pearce, 1986). Specifically, the sustained use of one claw enhances nerve activity, transforming fast-twitch muscle fibres into slow-twitch fibres, which leads to the development of a powerful crusher claw by influencing the attached exoskeleton. Meanwhile, reduced stimulation in the other claw preserves fast-twitch fibres, resulting in the formation of the more agile and precise cutter claw (Lang et al., 1978; Govind, 1992; Goldstein and Tlusty, 2003). Once this claw asymmetry is established, it becomes permanent, and even if a claw is lost later, it regenerates as the same type (Govind and Kent, 1982). This process highlights that while there is a genetic predisposition for claw specialization, it is the experience with specific stimuli during this key developmental period that determines whether a claw will become the crusher.

Suboptimal or stressful environments – such as standard rearing conditions lacking natural prey or appropriate substrates – can deprive lobsters of the necessary sensory stimulation, resulting in the development of symmetrical and less functional claws (i.e. double cutter claws; Govind, 1989). Such loss of functional asymmetry compromises lobsters' ability to capture food, defend against predators, and compete for resources (i.e. mating), thereby reducing their fitness and survival in the wild (Govind, 1989; Goldstein and Tlusty, 2003). These issues are particularly relevant in conservation programs where captive-reared lobsters are released into the wild to restore depleted populations or enhance wild stocks (Bell et al., 2008; Froehlich et al., 2017). In this context, it is essential for juvenile lobsters to possess, or rapidly acquire, phenotypic traits that closely resemble those observed in wild conspecifics (i.e. asymmetric claws; Daly et al., 2021). Thus, identifying the developmental stages that are more responsive to environmental cues and determining the specific stimuli required to promote wild-like phenotypic traits are critical steps for designing rearing protocols that not only improve the welfare of animals in

captivity but also enhance the success of conservation efforts (Zhang et al., 2022; 2023; Daly et al., 2021; Crates et al., 2023; Weber and Pinho, 2024).

Building on previous research conducted on the American lobster *H. americanus* (Govind and Pearce, 1989; Goldstein and Tlusty, 2003), which showed that early exposure to natural substrate (e.g. oyster shells) stimulates claw activity and the random development of the crusher claw, this study examines whether similar effects occur in the European lobster (*H. gammarus*), a decapod species that has been the focus of conservation programs for decades (Bannister and Addison, 1998; Ellis et al., 2015). Specifically, we investigated in a controlled setting how early-life environmental conditions, namely the presence or absence of physical enrichments, influence claw asymmetry and identified the key period when this effect is most pronounced. Two types of enrichment were considered: the first was a natural substrate composed of crushed oyster shells, previously shown to promote claw differentiation through mechanical stimulation in captive lobsters (Lang et al., 1978; Govind and Pearce, 1989), which also represents more natural conditions, mimicking the presence of substrate on the seafloor. The second type of enrichment was represented by the presence of an artificial shelter, which helps reduce stress (e.g. Poole, 1997; Olsson and Dahlborn, 2002; Zhou et al., 2023) and, by allowing the expression of basic innate behaviours (e.g. Poole, 1997), supports the development of appropriate antipredator behaviour during early growth stages (Van der Meeren, 2001; Carere et al., 2014). We hypothesized that the presence of environmental enrichments would increase claw asymmetry, and that the most sensitive period for this development would likely occur around the 5th benthic stage, when lobsters start to exhibit fully benthic behaviour and have ample opportunity to engage in claw use. By addressing these questions, this study advances our understanding of the mechanisms underlying developmental plasticity and offers valuable insights for optimizing conservation and reintroduction strategies.

## RESULTS

Out of the initial 398 individuals, 244 were included in the analysis (E=62; SH=58; SU=64; C=60) while 155 were excluded due to early mortality (before stage 4), missing claws or deformities, resulting in a total of 578 measurements (2.33±1.19 number of exuviae per individual). If individuals were alive throughout the experiment, measurements were taken at every stage, except in a few cases where the juveniles ate the exuvia before the daily control.

Overall, the analysis revealed that both the experimental treatments and the developmental stages, as well as their interaction, significantly influenced the process of claw differentiation, while mother identity showed no effect (Table 1). Juvenile lobsters reared with only substrate (SU group) exhibited significantly higher asymmetry scores compared to those in the

**Table 1. Results of linear mixed effects models (LMMs) and linear models (LMs) across developmental stages (4th to 7th) and treatments (fully enriched, substrate only, shelter only, control) in 244 individual European lobsters (*H. gammarus*)**

| | | Sum Sq | Mean Sq | NumDF | $F$ | Pr(>$F$) | $R_m$ | $R_c$ |
|---|---|---|---|---|---|---|---|---|
| *Full model* | Treatment | 365.69 | 40.63 | 9 | 5.54 | **<0.001** | 0.21 | 0.28 |
| | Stage | 321.89 | 107.30 | 3 | 14.63 | **<0.001** | | |
| | Treatment×stage | 510.46 | 18.91 | 27 | 2.58 | **<0.001** | | |
| *4th stage* | Treatment | 86.21 | 9.58 | 9 | 1.96 | **0.046** | - | - |
| *5th stage* | Treatment | 127.16 | 14.13 | 9 | 1.66 | 0.105 | - | - |
| *6th stage* | Treatment | 191.65 | 21.29 | 9 | 3.02 | **0.003** | - | - |
| *7th stage* | Treatment | 507.50 | 56.39 | 9 | 4.31 | **<0.001** | - | - |

The marginal $R^2$ ($R_m$) reflects variance explained by fixed factors while the conditional $R^2$ ($R_c$) accounts for both fixed and random effects. For statistically significant effects, *P*-values are highlighted in bold.

other treatment groups (E, SH, C; Fig. 1A,B; Table 1, Table S1). At the end of the experiment, juveniles raised with substrate (SU group) from stage 4th, 5th or 6th, were the only ones to exhibit asymmetry coefficients that exceeded the 5% threshold (Fig. 1). Individuals belonging to the other experimental groups (E, SH, C) displayed comparable asymmetry levels (Fig. 1B, Table S1). The specific developmental stages at which physical enrichments were introduced into the rearing compartments – namely stages 4th, 5th or 6th – did not significantly alter the trajectory of claw differentiation in any of the experimental groups (Table S1).

Regarding ontogenetic changes, no significant effects on claw asymmetry rates were observed across treatments between the 4th and 5th benthic stages (Table S1). However, after the 6th stage, a strong increase in asymmetry rates was recorded in the three treatments belonging to the substrate group, which significantly differed from the other experimental conditions (Table S1). During

this period, no increase in asymmetry was noted in the shelter, shelter and substrate, and control groups (Fig. 1A).

Throughout the experimental phase, the group reared with only substrate exhibited lower mortality rates (33.3% of mortality) compared to the other experimental conditions, which instead showed similar values among them (E=46%; SH=45.4%; C=48%).

## DISCUSSION

Our analysis provides clear evidence that early-life experience, specifically exposure to the substrate alone without additional physical enrichments, significantly influences the development of asymmetric claws, a key ecological trait in the European lobster (Govind, 1989; Goldstein and Tlusty, 2003; Daly et al., 2021).

The addition of oyster shells to rearing compartments – introduced at the 4th, 5th or 6th stage – was highly effective in promoting claw differentiation. While the substrate group showed an average gradual increase in asymmetry rates as development progressed, the other enriched conditions yielded similar results (Fig. 1A). This effect became particularly pronounced between the 6th and 7th benthic stages, coinciding with the transition of juveniles to a fully benthic lifestyle (Cobb et al., 1971; Castro and Cobb, 2005). These findings highlight the critical influence of environmental stimuli in directing phenotypic development through epigenetic mechanisms, with meaningful implications for both conservation strategies and animal welfare.

Notably, lobsters reared with substrate alone exhibited higher asymmetry scores compared to those provided with a combination of shelter and substrate, shelter only, or no enrichment. These results, based on the introduction of various physical enrichments in a controlled setting, enhance existing research on clawed lobsters and confirm that substrate availability encourages selective claw use, ultimately promoting bilateral asymmetry (Govind and Pearce, 1986, 1989; Goldstein and Tlusty, 2003). However, the effect of substrate can be influenced by the presence of other environmental factors. For instance, lobsters reared with both substrate and shelter exhibited lower asymmetry scores than those with substrate only. This difference may be attributed to reduced interaction with the substrate, as juveniles likely spent more time inside the shelter. This idea is supported by previous studies, which have shown that starting from the 5th benthic stage, wild juvenile lobsters display stronger benthic behaviour, progressively spending more time in shelters as they age and thus limiting other activities such as substrate manipulation (Botero and Atema, 1982; Cobb, 1971; Castro and Cobb, 2005). However, in natural environments, wild lobsters making use of natural shelters retain the possibility of interacting with physical stimuli (like substrate, pebbles, or food items), and the increased use of a shelter would not hinder claw development. To test this hypothesis, future studies should provide a fully enriched condition where oyster shells can be placed both inside and outside the shelter. This arrangement would allow the juveniles to interact with the substrate while still having access to the shelter, potentially stimulating claw differentiation.

The timing of the introduction of the physical stimuli into the rearing compartments did not significantly affect claw differentiation in any of the enriched groups. Specifically, in the substrate group, which had the most noticeable impact on claw differentiation, adding oyster shells at stages 4th, 5th or 6th resulted in similar asymmetry levels. This contrasts with the findings of Govind and Pearce on *H. americanus* (1989), who observed that the presence of substrate at the 5th stage alone caused bilateral asymmetry in the claws, while the substrate at the 6th stage had

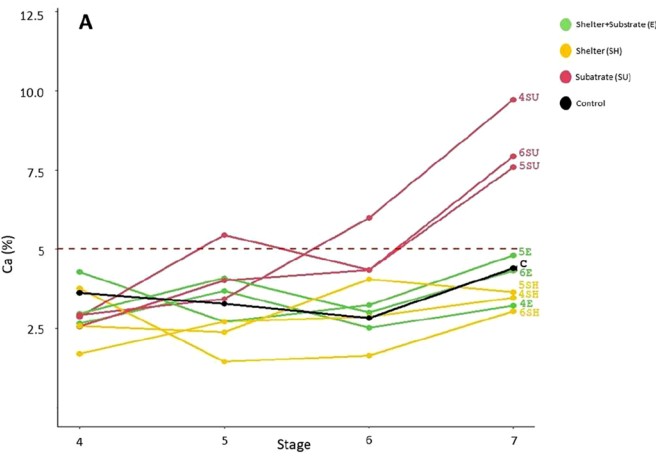

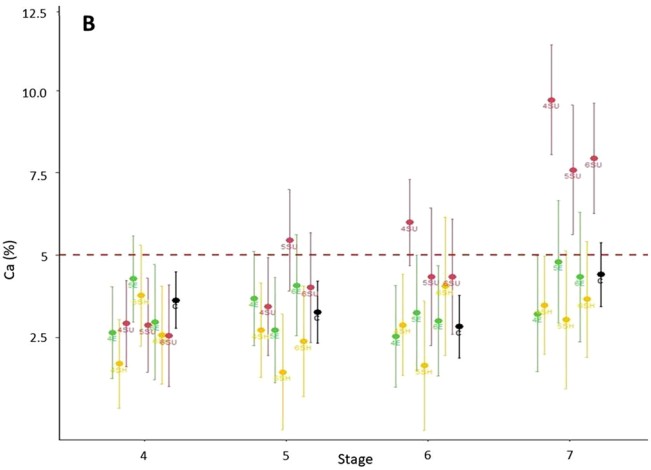

**Fig. 1. Claw asymmetry coefficient (Ca) values for 4th, 5th, 6th and 7th stage lobsters under varying rearing conditions.** The study involved 244 juvenile lobsters (*H. gammarus*) exposed to four different enrichment conditions provided during their first three benthic stages (4h, 5th, 6th): full enrichment (E, green; *N*=62) with both a shelter and substrate, partial enrichment with only shelter (SH, orange; *N*=58), partial enrichment with only substrate (SU, red; *N*=64), and control without any enrichment (C, black; *N*=60). In panel A, trend lines illustrate how Ca changes over time across the different treatments. Panel B highlights the mean Ca values with corresponding standard deviations for each treatment at each development stage. The red dashed line represents the threshold value distinguishing symmetry (<5%) from asymmetry (>5%) in the claws.

no effect. The differences between these studies may be due to variations in the biological characteristics of the species, differences in the rearing conditions, or factors related to species' origin and the environment they come from. Nonetheless, it is evident that for both American and European species, the earliest benthic stages represent the most critical period during which the central nervous system (CNS) is susceptible to lateralization, leading to the differentiation of a crusher and a cutter claw (Govind, 1992). In contrast, although not directly tested in this experiment, it is probable that the CNS loses its plasticity in later developmental stages, thereby limiting its ability to induce claw differentiation as development progresses (Emmel et al., 1908; Lnenicka et al., 1988; Govind and Pearce, 1989).

Regarding developmental stages, a gradual increase in asymmetry rates, particularly in the substrate group, was observed starting from the 6th stage. In the earlier developmental stages, specifically the 4th and 5th benthic stages, no differences in asymmetry rates across the various experimental treatments could be detected. This finding can be explained by the fact that claw differentiation during these stages is still in its early phase, primarily driven by neurological factors, and has not yet led to visible morphological changes influenced by environmental interactions (Govind and Pearce, 1989; Govind, 1992). In addition, under natural conditions, juveniles in the 4th benthic stage have not yet developed a fully benthic behaviour, which begins to emerge between the 5th and 6th benthic stages, when they start exploring the seafloor and transitioning to a bottom-dwelling lifestyle (Botero and Atema, 1982; Cobb, 1971; Castro and Cobb, 2005). For these reasons, the effects of substrate presence on asymmetry rates may not be immediately apparent in these early benthic stages. However, as development progresses and juveniles increase the interactions with the substrate, neurological processes begin to influence muscle fibres composition, determining which claw will develop as the 'crusher' and which as the 'cutter' (Lang et al., 1978; Govind, 1992; Goldstein and Tlusty, 2003). Notably, asymmetry values in individuals from the substrate group begin to slightly diverge from those in other experimental groups between the 5th and 6th stages. This divergence becomes more pronounced between the 6th and 7th stages, as early neurological differences likely manifest in muscle composition and claw morphology, leading to significantly higher asymmetry rates in the substrate group compared to the other experimental groups.

Building on these results providing appropriate environmental stimuli during the sensitive developmental period becomes essential to ensure a wild-like development of captive reared animals and thus for enhancing the effectiveness of conservation programs (Näslund, 2021; Daly et al., 2021). We therefore recommend enriching the hatchery environment with oyster shells starting from the 4th, 5th or 6th benthic stage and ensuring that other types of enrichments, such as shelters, are designed to avoid interferences with the possibility of interacting with the substrate. For instance, shelters could have a half-conic shape or consist of a low-hanging cover of one of the cell's corners. Additionally, to further enhance conservation efforts, it is also essential to carefully plan the exact timing of reintroduction activities. While releasing juveniles too early can result in mortality rates similar to those experienced in the wild (Van der Meeren, 1993, 2000, 2005), keeping them in captivity too long not only increases maintenance costs for humans, but may also fosters traits adapted to controlled environments, diminishing their ability to thrive in natural conditions (Pigliucci et al., 2005; Hutchings and Fraser, 2008; Chittenden et al., 2010). To address this trade-off, we recommend releasing juvenile lobsters at or shortly after the 6th

developmental stage, as they are more likely to exhibit appropriate benthic behaviour (Botero and Atema, 1982; Cobb, 1971; Castro and Cobb, 2005), well-developed claw musculature, and thus proper asymmetry.

It is worth emphasizing that all these considerations are not only fundamental for conservation activities but also for ensuring a certain level of animal welfare in captivity. Meeting the needs of captive reared animals is both an ethical responsibility and a cornerstone for the success of rearing programs (Chiang and Franks, 2024). Providing appropriate rearing conditions that support natural development, such as physical enrichments, can minimize stress, reduce abnormal behaviours, and improve the overall quality of reared organisms (Näslund and Johnsson, 2016; Barreto et al., 2021). Such measures also help to minimize the occurrence of deformities and infections, improving individual health and subsequently lowering mortality rates (Webster, 1995; Zhang et al., 2022). These aspects may explain the lower mortality rates observed in the substrate group, which likely experienced reduced stress levels compared to conspecifics belonging to the other experimental conditions. Therefore, attention to welfare conditions in rearing protocols may help to increase the biological and behavioural quality of reintroduced organisms, ultimately increasing the value and sustainability of conservation initiatives (Näslund and Johnsson, 2016; Zhang et al., 2022; Zhang et al., 2023).

While our findings highlight the significant role of environmental factors in driving the development of claw asymmetry, the underlying molecular pathways remain insufficiently understood. To address this gap, future research should aim to elucidate the epigenetic mechanisms that mediate these developmental processes. Advanced techniques, such as transcriptomic analyses and DNA methylation profiling, could be used to examine gene expression shifts during the critical period of claw differentiation, offering valuable insights into how environmental stimuli regulate genomic activity (Tammen et al., 2013; Al Aboud et al., 2023). Such research would clarify the complex relationship between genetic programming and environmental inputs in shaping phenotypic traits. By addressing these gaps, we can better integrate scientific understanding into practical applications, advancing sustainable and ethical practices in conservation and aquaculture.

## MATERIALS AND METHODS
### Study species
The European lobster is a decapod crustacean that inhabits the Atlantic coast of Europe and much of the Mediterranean basin. This species has a complex life cycle that starts with fertilized eggs carried by the female for 9-12 months. After hatching, the larvae pass through three pelagic stages, lasting about 4 weeks, where they float freely in the water column. These stages are characterized by molting and rapid growth (Nichols and Lawton, 1978; Charmantier et al., 1991). After the third larval molt, they enter the post-larval or juvenile (4th) stage and start to become benthic (bottom-dwelling). From the 5th stage, particularly in the 6th and 7th stages, lobsters adopt a fully benthic lifestyle and morphology, resembling small adult lobsters, and continue to grow through periodic molting (Charmantier et al., 1991; Govind, 1992).

### Berried females and larval rearing
From August to November 2023, five ovigerous females were caught by local fishermen from two distinct locations; one on the Italian west coast (−42.327235, 11.572900) and the other four from the Italian north-east coast (45.257167, 12.727861). These females were transported to the Ichthyogenic Experimental Marine Centre (CISMAR, Tarquinia – VT, Italy −42.201851, 11.721749) and housed in individual 1500 l culture tanks with a 1.5 l water turn-over rate connected to a Recirculating Aquaculture System (RAS), until egg hatching. Water flowed through a

suspended solid removal unit (63 µm mesh), a moving bed bio-filter reactor, a foam fractionator and a UV sterilizer before flowing back to the holding tanks. Water temperature was set at 17°C. Salinity, dissolved oxygen, pH and nitrogen compounds (TAN; $NO_2^-$; $NO_3^-$) were monitored daily and kept within their optimal range ($\sim$38‰; $\sim$9.5 mg l$^{-1}$; $\sim$8.4; <1.0 mg l$^{-1}$; <0.4 mg l$^{-1}$; <20 mg l$^{-1}$). The lighting schedule was maintained on a semi-natural dark-light cycle (8:30-17:30) with lamps (36 W) controlled by a timer and complemented with the natural light spreading through the large windows of the laboratory facility. During the 3-4 days of hatching, the females were individually transferred to 74 l birthing chambers filled with the same seawater of the aforementioned RAS. After this period, the females were returned to their original tanks. Newly-hatched larvae were collected every morning, counted, and housed in 200 l upwelling vessel with heavy aeration over the entire planktonic phase. The upwellers were connected to the same RAS of the adults. Water flow was set at 300 l h$^{-1}$ to guarantee a 1.5 water turnover rate. Planktonic larvae (stages 1 to 3) were stocked at 20 larvae l$^{-1}$ (Hughes et al., 1974; Wickins et al., 1995; Bell et al., 2005) and fed *ad libitum* twice a day with a mix of frozen *Artemia* sp., *Mysis* sp. and krill (*Euphasiidae* spp.). Benthic larvae (from 4th stage onward) were provided daily with a specialised protein concentrate formulated for crustaceans' larvae (55% crude protein, 15%% lipid, 5% fibre, 13% ash; Aquahive® feed, https://www.oceanonland.com/). To guarantee a varied diet, benthic larvae were also supplemented twice a week with the same crustacean mix used during the planktonic phase. Water samples taken from each tank were analysed weekly for levels of ammonia, nitrate, pH, and dissolved oxygen. The holding spaces of the benthic larvae (described in detail below) were siphoned and cleaned daily to minimize the accumulation of solid organic matters and waste products.

## Experimental procedure

Once the larvae reached the benthic phase (4th stage), approximately 2 weeks from hatching, a total of 398 juveniles were equally and randomly divided in four floating grids, composed by 100 square individual compartments (8 cm per side, 3 cm deep). Each grid was located in a different 1500 l tank. The four treatments were spread equally across the different 1500 l tanks (Fig. 2). Experimental groups were defined by the presence/absence and composition of physical enrichment provided in the individual compartments (Fig. 2). Four distinct experimental conditions were implemented: (1) partial enrichment with substrate (SU; $N$=99) made of oyster shells (*Magallana gigas* spat, 0.5-1 mm), (2) partial enrichment with shelter (SH; $N$=99) composed by a PVC tube (3 cm in length, 2 cm in diameter), (3) a full enrichment with a combination of both substrate and shelter (E; $N$=100) and (4) a control group, in which none of the aforementioned enrichments was provided (C, control; $N$=100). Physical enrichments were introduced at specific developmental stages, namely the 4th, 5th and 6th benthic stages, in order to identify a potential sensitive period for claw development. Following this procedure a total of 10 treatments were obtained: 4SU, 5SU, 6SU, 4E, 5E, 6E, 4SH, 5SH, 6SH, C.

## Morphometric measurements

Rearing compartments were checked daily to collect exuviae (stored in 1.5 ml Eppendorf containing 70% diluted alcohol), which were used to measure morphological parameters. Juveniles that showed deformed ($\sim$3.1% of all measured exuviae) or absent claws ($\sim$9.1% of all measured exuviae) were excluded from the sample. To determine the most critical period for the development of bilateral asymmetry, claw length and width were monitored during the first four stages of benthic development. Exuviae from each treatment were collected during the 4th, 5th, 6th and 7th stages, and photographed using a Leica M205 FCA stereomicroscope connected to a PC equipped with Leica Application Suite X (LAS X) software, which enabled precise measurement of claw length and width for each individual. To measure claw length, we considered the longest point from the proximal end of the claw to the distal tip, while for the width we considered the span of a line perpendicular to the length line at the point where the dactyl joins the pollex (Fig. 3). The values of length (L) and width (W) were used to compute a L/W ratio for each claw. The claw with the larger L/W was identified as the cutter and the claw with the smaller L/W ratio was identified as the crusher. Claw asymmetry (Ca) was then calculated as a coefficient, following the formula proposed by Goldstein and Tlusty (2003):

$$Ca = (1 - (Rr/Ru)) \cdot 100.$$

If the asymmetry coefficient was less than 5%, the individual was considered symmetrical, whereas if the asymmetry was equal to or exceeded

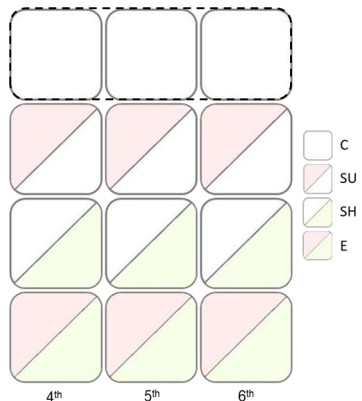

**Fig. 2. Representation of the experimental procedure.** A total of 398 juveniles were randomly divided between four distinct enriched conditions, introduced at specific developmental stages – namely the 4th, 5th, and 6th benthic stages – resulting in a total of ten treatments were obtained: 4SU, 5SU, 6SU, 4E, 5E, 6E, 4SH, 5SH, 6SH, C. The treatments were: (i) partial enrichment with substrate (SU; N = 99) made of oyster shells (*Magallana gigas* spat, 0.5-1 mm), (ii) partial enrichment with shelter (SH; N = 99) composed by a PVC tube (3 cm in length, 2 cm in diameter), (iii) a full enrichment with a combination of both substrate and shelter (E; N = 100) and (iv) a control group, in which none of the aforementioned enrichments was provided (C; N = 100). Each tank hosted a grid arranged as illustrated, with the four treatments equally represented in each grid and tank.

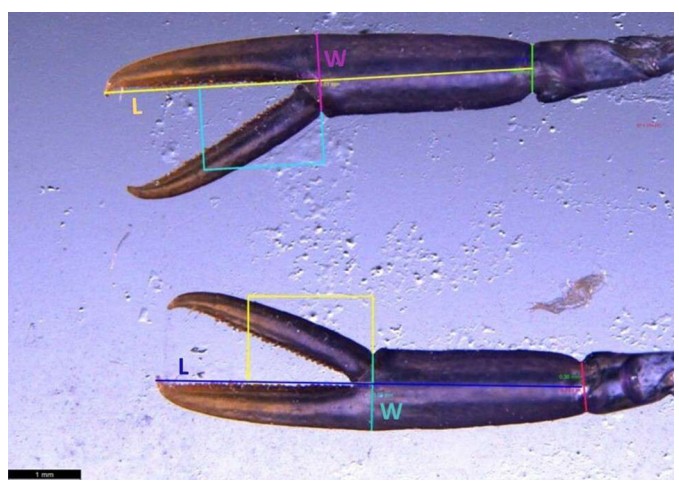

**Fig. 3. Claw morphology measurements under stereomicroscopic imaging.** The image illustrates the measurement of claw length (L) and claw width (W) from collected exuviae. Length (L) was defined as the longest distance from the proximal end to the distal tip of the claw, while width (W) represents the span perpendicular to the length at the junction of the dactyl and pollex. These metrics were used to calculate the L/W ratio, identifying the cutter claw (higher ratio, Ru) and the crusher claw (lower ratio, Rr). Measurements were captured using a Leica M205 FCA stereomicroscope and Leica Application Suite X software.

5%, the individual was considered asymmetrical (Goldstein and Tlusty, 2003).

## Statistical analysis

We were interested in determining whether juveniles exposed to varying degrees of environmental complexity during different stages of early benthic development differed in claw asymmetry rates. Accordingly, we fitted linear mixed-effects models with claw asymmetry coefficient (Ca) as the dependent variable. We included treatment (4E, 5E, 6E, 4SH, 5SH, 6SH, 4SU, 5SU, 6SU, C), developmental stage (4th to 7th), and their interaction as fixed effects. Individual identities nested within the Mother ID were incorporated in the random structure of each model (random intercepts) to account for repeated measures, as well as possible genetic effects.

Analyses were first performed on the whole dataset; since we found an interaction between the explanatory variables treatment and developmental stage, we ran post hoc analyses on subsets of data including only one developmental stage at a time. Due to limited sample sizes in these subsets, it was necessary to use linear models rather than linear mixed-effects models. We used default families in the mixed models and verified the normality of the weighted residuals by using quantile-quantile plots and lines crossing the first and third quartile, although linear mixed models are robust to violations in distributional assumptions (Schielzeth et al., 2020). The significance level was set at $\alpha < 0.05$. Data analysis was performed with R. Studio v. 4.2.1 (R Core Team, 2016), using the packages *lmerTest* and *Emmeans* (Kuznetsova et al., 2013; Lenth, 2018).

## Acknowledgements

We thank Doriana Benedetti and Mirco Liuzzo for helping in lobsters maintenance and during data collection, and all the CISMAR colleagues who contributed to support the experiment. Furthermore, we would like to thank the University of Tuscia for providing the facilities and equipment of CISMAR, and the University of Florence and the University of Venice for their logistical support.

## Competing interests

The authors declare no competing or financial interests.

## Author contributions

Conceptualization: L.L., G.G., E.B., D.C., C.C.; Data curation: L.L., G.B., R.D.D.; Formal analysis: L.L.; Funding acquisition: C.C.; Investigation: L.L., G.B., G.G., R.D.D.; Methodology: L.L., G.B., G.G., R.D.D.; Project administration: E.T., S.M., D.C., C.C.; Resources: E.T., S.M., G.N., D.C., C.C.; Validation: V.M., D.C., C.C.; Writing – original draft: L.L.; Writing – review & editing: L.L., G.B., V.M., R.D.D., E.T., S.M., D.C., C.C.

## Funding

The study was supported by the PRIN project PL-ASTICI (20223EETLW) 'Phenotypic plasticity in a rapidly changing world: an ontogenetic perspective for improving aquaculture and conservation practices of the European lobster' granted to C.C. (PI), S.M., and E.T. and was part of the master's thesis of G.B. in Marine Ecology and Biology, at the University of Tuscia. The project was also implemented under the National Recovery and Resilience Plan (NRRP), Mission 4 Component 2 Investment 1.4 - Call for tender number 3138 of 16 December 2021, rectified by Decree number 3175 of 18 December 2021 of the Italian Ministry of University and Research funded by the European Union - Next Generation EU. Project code CN_00000033, Concession Decree number 1034 of 17 June 2022 adopted by the Italian Ministry of University and Research, CUP J83C22000860007, Project title 'National Biodiversity Future Centre -NBFC'; and Mission 4 Component 2 Investment 3.1. - Italian Ministry of University and Research funded by the European Union – NextGenerationEU; Project code IR0000035, CUP C63C22000570001, Project title 'Unlocking the Potential for health and food from the seas' - EMBRC UP. During manuscript preparation, V.M. was supported by the National Biodiversity Future Center—NBFC, a project funded by the European Union 'NextGenerationEU' (National Recovery and Resilience Plan (NRRP), Mission 4 Component 2 Investment 1.4—Call for tender number 3138 of 16 December 2021, rectified by Decree number 3175 of 18 December 2021 of Italian Ministry of University and Research funded by the European Union—Next Generation EU. Project code CN_00000033, Concession Decree number 1034 of 17 June 2022 adopted by the Italian Ministry of University and Research, CUP J83C22000860007). Open Access funding provided by University of Florence. Deposited in PMC for immediate release.

## Data accessibility

Data and the R code used for the statistical analysis and figures preparation of this work are available at https://doi.org/10.6084/m9.figshare.28184723.v1.

## Peer review history

The peer review history is available online at https://journals.biologists.com/bio/lookup/doi/10.1242/bio.061901.reviewer-comments.pdf

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
