## [Peer Review File · Biology Open]

Early-life environment shapes claw bilateral asymmetry in the European lobster (*Homarus gammarus*)

Lorenzo Latini, Gioia Burini, Valeria Mazza, Giacomo Grignani, Riccardo De Donno, Eleonora Bello, Elena Tricarico, Stefano Malavasi, Giuseppe Nascetti, Daniele Canestrelli, Claudio Carere

DOI: 10.1242/bio.061901

Editor: Lewis Halsey

Review timeline

Original submission:	17 January 2025
Editorial decision:	24 January 2025
First revision received:	3 February 2025
Accepted:	5 February 2025

Original submission

First decision letter

MS ID#: bio.061901

MS TITLE: Early-life environment shapes claw bilateral asymmetry in the European lobster (*Homarus gammarus*)

AUTHORS: Lorenzo Latini; Gioia Burini; Valeria Mazza; Giacomo Grignani; Riccardo De Donno; Eleonora Bello; Elena Tricarico; Stefano Malavasi; Giuseppe Nascetti; Daniele Canestrelli; Claudio Carere

I have now reached a decision on the above manuscript.

The reviewer reports are shown at the bottom of this email or can be accessed, together with a copy of this decision letter, by going to:

As you will see, the reviewers gave favourable reports, but raised some critical points that will require amendments to your manuscript. I hope that you will be able to carry these out, because we would like to be able to accept your paper.

Please ensure that you clearly highlight all changes made in the revised manuscript. I should be grateful if you would also provide a point-by-point response detailing how you have dealt with the points raised by the reviewers in the 'Response to Reviewers' box. Please attend to all of the reviewers' comments. If you do not agree with any of their criticisms or suggestions please explain clearly why this is so.

Reviewer 1

Comments for the author

General comments:

The manuscript described laboratory experiments on lobsters through their early life phases to assess the impact that the environment has on phenotype, namely asymmetry in claws. Authors

explain that these lobsters require asymmetrical claws to optimise fitness in the wild, as the two different claw sizes act as different 'tools' (i.e. crushing vs cutting/pinching). The authors conducted an appropriate experiment and analysis to assess the question of whether environment affects asymmetrical claw development, with reasonably intuitive results illustrating that high levels of substrate are needed to develop asymmetry in claws. There is an interesting result that the treatment with half substrate + shelter did not produce asymmetry (only full substrate without shelter caused asymmetry). The study is relevant to rearing the species suitably in captivity for release in the wild, as well as for more general animal ethics concerns of captive animals. It's a nice study, well written, and simply presented, so it was easy to read. I have only minor comments and suggestions for the authors which I hope will improve their manuscript. The main issues I see are in the presentation of the methods, which requires some clarification in parts. However these are relatively minor and I believe are oversights in writing, rather than errors or problems in the experimental methodology itself.

Specific comments:

Ln 37-39. But exposure was not enough - they were exposed to substrate in the substrate+shelter group and the asymmetry didn't develop. I think the abstract can be clearer on this point.

Ln 49. Change "fit" to "fitness".

Ln 51. Change "population" to "populations".

Ln 89. Here you call it the "pincer" claw but the rest of the manuscript calls it the 'cutter' claw - please be consistent.

Ln 120. Remove "very" before "complex".

Ln 123-126. It's not clear to me what the difference between the 4th and 5th stage are? Are they only partially benthic in the 4th stage?

Ln 128. The start of this sentence is unclear, perhaps it's better to start with "From (month) to (month) 2023, five ovigerous..."

Ln 128-130. Were there any differences in the results between locations the mothers were collected from? e.g. did the offspring from the west coast mother have different Ca values?

Ln 132. Abbreviation (RAS) should come second after the term and be in parentheses.

Ln 140. Does "they" refer to the mothers?

Ln 150. "Grids" haven't been mentioned - do you mean 'tanks'?

Ln 153. Missing "the larvae" after "once".

Ln 155. Were the four treatments spread across the four different 1500L tanks? e.g. if there was a pathogen present in only one of the tanks could that have affected the outcomes if only one treatment occupied each tank?

Ln 167-168. The use of "ca" is slightly confusing here because it's the name of your main variable of interest which is also a percentage unit.

Ln 175-177. I had to read this sentence a couple of times to understand - can you please clarify that this means the claw with the larger L/W was the cutter and the claw with the smaller L/W ratio was the crusher.

Ln 180-181. I'm being pedantic, but what about exactly 5%?

Ln 192. Change "model" to "models".

Ln 192-193. Was this a Tukey's test? The table caption says Tukey's but here says Bonferroni, can you please clarify. The public R code provided (well done!) suggests it was Tukey's not Bonferroni.

Ln 200-201. Were there any differences among groups for the number of excluded lobsters? I.e. did the treatments have an effect on the rate of missing or deformed claws?

Ln 202. I assume this is because of logistical/practical reasons, but can you explain why measurements weren't taken at every stage?

Ln 217. Please name the other experimental groups.

Ln 237-241. I think this information is key background knowledge to the study and its underlying assumptions about mechanism of action, so I think it should be moved to the introduction.

Ln 263. Change "claws" to "claw".

Ln 284 - 294. This is a bit repetitive of previous text - it sounds like a conclusion but then more paragraphs follow. I think this could be shortened and moved to the final paragraph or just cut from the manuscript.

Ln 304. Make it clear that this is maintenance costs for humans, not the lobsters.

Figure 2 - can you add a marker or dashed horizontal line at $y = 5\%$ to indicate the threshold of asymmetry.

Table 2- I think this can be moved to supplementary material.

Reviewer's Responses to Questions

Experimental Quality

Does each figure have the proper controls?

Yes

Are experiments performed using appropriate methods that will answer the question (or test the hypothesis or support the observations) posed by the authors? Is the right tool used for the job?

Yes, yes.

Were the data analyzed using appropriate statistical tests?

Yes but some clarity needed.

Reproducibility

Were experiments in each figure performed using adequate number of biological replicates?

No power analyses were presented but I would judge yes for the main treatment effects (substrate, shelter, mixed and control). The subset treatments (interactions with life phase) may have benefitted from larger sample size, but the authors acknowledge this and perform appropriate statistical tests, and additionally I don't see any evidence that increased sample size would change the conclusions.

Is there sufficient raw data to assess the rigor of the analysis?

Yes, data and code were uploaded to a public repository.

Does the methods section provide sufficient detail to permit reproducibility?

No - some minor clarifications are needed (see comments below).

Completeness

Are the author's conclusions supported by the data?

Yes.

Are there any flaws in the experimental design that invalidate the approach taken by the authors?

No.

Are there experiments that have not been performed, but if true would disprove the conclusion? If yes, and if such experiments would be costly or time-consuming to perform, do the authors acknowledge this in a discussion of the limitations?

No, but there are further experiments needed to unpack the fact that an effect was observed for substrate-only treatment but not substrate+shelter treatment. The authors acknowledge this, suggest an explanation, and suggest future experiments to tackle this question.

Scholarship

Do the authors cite and discuss the merits of relevant data that would argue against their conclusion?

Yes.

Do the authors cite and discuss the merits of relevant data that would support their conclusion?

Yes, the work is heavily based on past work.

Reviewer 2

Comments for the author

The authors conducted a randomized controlled trial, whereby a group of juvenile European lobster were randomized to one of 4 experimental conditions (full enrichment comprising the provision of both substrate and shelter; partial enrichment with substrate only; partial enrichment with shelter only and a control condition with no environmental enrichment). The enrichments were introduced at different developmental stages (i.e., the 4th, 5th and 6th benthic stages). The development of claw asymmetry - a critical factor for lobster fitness - was the primary outcome of interest. The

results indicated that substrate provision, irrespective of stage of introduction encouraged greater claw asymmetry, and this differentiation was primarily evident from the 6th stage on.

Overall, I enjoyed this article. The biological question is relevant and interesting, the study was well-designed and the interpretation, both in terms of scientific and practical relevance was compelling. Both the design and the statistical analysis are quite complex, and there were some aspects that I believe require further clarification and contextualization. I have made a number of suggestions below which I hope will be of use to the authors.

Justification of the chosen enrichments:

It would be useful to expand the explanation of the chosen enrichments, and their relevance to both captive and wild-living lobsters. What is the purpose of providing the substrate in the form of oyster shells? Is it because gripping these shells provides the mechanical stimulation necessary to develop claw asymmetry, or do they contain some nutrient that was not available to the juvenile lobsters that were fed the standard diet described on lines 145 - 146?

I understand that in the wild shelter may be important to protect against predation, but in a controlled environment, what potential benefit does enriching the environment with shelter offer? Perhaps reduced stress levels by providing a place to hide?

Statistical Analysis:

I found the statistical analysis section somewhat difficult to follow, and suggest that some re-working to enhance clarity may be useful. I write this under the caveat that I am not a statistical expert, and I recognize that this is a complex analysis with many potential applications. As such, I consider these points as suggestions for your consideration rather than as requests.

1. Line 191 - 192. The justification for conducting isolated, in addition to the full model was not clear to me. How did this approach overcome the limitations of the small sample size available in different subsets? Indeed separating points in time could potentially dilute the overall treatment effect of certain conditions, e.g., there is no way that enrichments introduced at stages 5 or 6 could influence timepoint 4 so why test for this? Furthermore, based on the information presented in Table 2, at the 5th stage the treatment effect was non-significant, and at the 4th stage it was pretty close to the pre-defined cut off of 0.05, whereas at the 6th and 7th stage the data are more convincing. This makes sense seeing as theoretically the enrichments would have more time to exert their influence at these later developmental stages. My suggestion would be to work from the full model, and to test specific hypotheses of interest (see below comment 4 about multiple comparisons) based on the results of this model along with the theoretical justification for the study.

2. I suggest that you include confidence intervals in Figure 3, Panel A. I realise that this will make the figure somewhat messy considering the number of conditions, however in my opinion this graph is the most informative representation of the overall results, but currently the lack of confidence intervals make it difficult to interpret. The dots and error bars could be dodged slightly to allow you to view different conditions that potentially overlap. An alternative suggestion would be to provide the numerical data with confidence intervals in a supplementary file.

3. How were the trend lines for Figure 3, Panel A calculated? Were they estimated from the R emmeans output? If so, the confidence intervals should also be automatically available within the output.

4. Table 2: Is it really necessary to include so many pairwise comparisons? Considering that you used a post-hoc adjustment, the sheer volume of comparisons likely means that you had very little power to detect potential differences in each one, but some of these comparisons seem unnecessary. For example, there is no need for all comparisons at earlier developmental stages if the enrichments were only offered later. My suggestion would be to theoretically justify the most relevant pairwise comparisons, based on your hypotheses and the outcomes of the full model, rather than to unnecessarily lose power by running all potential comparisons.

A potential complementary analysis could be to conduct an area under the curve analysis, which may overcome some of the limitations of adjusting for multiple comparisons and may provide a clearer indication of which treatment led to the greatest gain in claw asymmetry. To be clear, I am not suggesting this as a replacement for the main model, but as a potential complement.

General Comments:

Lines 28 - 29 and 48 - 49: The definition provided here seems more appropriate to describe plasticity more generally, whereas I would consider developmental plasticity to be specific to the early-life environment, and how this can shape later-life outcomes.

Lines 407 - 408: These studies by Govind & Pearce, 1989 and Goldstein et al. 2003 appear fundamental to the current one, so it would be useful to more explicitly describe their findings and to explain how the current study builds on these.

Line 125 - 126: Considering that these developmental stages are core to the study design and interpretation, I suggest that you explicitly define stages 6 and 7 at this point. I assume that these refer to the phases of periodic molting mentioned on line 125? Are the lobsters considered mature at stage 7? It may also be interesting to include the developmental stage in brackets when you later refer to different feeding regimens (Lines 144 - 148). I assume that the planktonic larvae are at stages 1 - 3 and the benthic larvae at stages 4 onward?

Line 153: Small typo here - should be "once the benthic phase was reached" or "once the juveniles reached the benthic phase".

Line 193: Table 2 indicates that a Tukey, not Bonferonni, adjustment was performed. Which was it?

Line 222: It is not accurate to state that "exposure to physical enrichments significantly influences the development of asymmetric claws" considering that one of the provided enrichments, i.e., shelter, had no influence, and indeed may actually have been counter-productive if it prevented the juvenile lobsters from interacting more with the substrate. I suggest re-phrasing this line to more accurately reflect the main findings.

Figure 1: Personally, I believe this Figure is unnecessary as it doesn't seem to add anything beyond what was described in the text. It could even be mis-leading as it implies that there are 12 groups rather than 10.

Reviewer's Responses to Questions

Experimental quality

Does each figure have the proper controls?

If 'No', please indicate reasons in Comments for Author box below.

Yes

Were the data analyzed using appropriate statistical tests?

If 'No', please indicate reasons in Comments for Author box below.

Yes

Reproducibility

Were experiments performed using adequate number of biological replicates?

If 'No', please indicate reasons in Comments for Author box below.

Yes

Does the methods section provide sufficient detail to permit reproducibility?

If 'No', please indicate reasons in Comments for Author box below.

Yes

Completeness

Are the manuscript's conclusions supported by the data?

If 'No', please indicate reasons in Comments for Author box below.

Yes

Scholarship

Do the authors cite and discuss the merits of data that would argue for and against their conclusion?

If 'No', please indicate reasons in Comments for Author box below.

Yes

Does the manuscript title & abstract accurately reflect the contents of the manuscript, without hyperbole?

If 'No', please indicate reasons in Comments for Author box below.

Yes

First revision

Author response to reviewers' comments

thank you for the chance to improve our manuscript. We are extremely grateful for your insightful and constructive comments, and we have revised our manuscript accordingly, also taking care it complies with the provided formatting guidelines. Please find below our revision letter addressing your comments point by point, and a revised manuscript with the revised section highlighted in yellow as an additional file. Here, we have included a detailed explanation of how we have dealt with each comment. Reviewers' comments are highlighted in italics, and our answers are in bold, green font and preceded by an asterisk.

Reviewers' comments

**Reviewer
#1**

General comments:

The manuscript described laboratory experiments on lobsters through their early life phases to assess the impact that the environment has on phenotype, namely asymmetry in claws. Authors explain that these lobsters require asymmetrical claws to optimise fitness in the wild, as the two different claw sizes act as different 'tools' (i.e. crushing vs cutting/pinching). The authors conducted an appropriate experiment and analysis to assess the question of whether environment affects asymmetrical claw development, with reasonably intuitive results illustrating that high levels of substrate are needed to develop asymmetry in claws. There is an interesting result that the treatment with half substrate + shelter did not produce asymmetry (only full substrate without shelter caused asymmetry). The study is relevant to rearing the species suitably in captivity for release in the wild, as well as for more general animal ethics concerns of captive animals. It's a nice study, well written, and simply presented, so it was easy to read. I have only minor comments and suggestions for the authors which I hope will improve their manuscript. The main issues I see are in the presentation of the methods, which requires some clarification in parts. However these are relatively minor and I believe are oversights in writing, rather than errors or problems in the experimental methodology itself.

* Thank you very much! We are very grateful for your appreciation, and for your helpful and insightful comments. We have revised the manuscript accordingly, with special attention to the methods presentation. We answer in detail to each of your comments below.

Specific comments:

Ln 37-39. *But exposure was not enough - they were exposed to substrate in the substrate+shelter group and the asymmetry didn't develop. I think the abstract can be clearer on this point.*

* Thank you for noticing. We clarified this aspect of the abstract, by adding the specification that it was the substrate alone, without other stimuli, to promote the development of the asymmetry (line 38).

Ln 49. *Change "fit" to "fitness".*

* Done (line 50). We removed the term "fitness".

Ln 51. *Change "population" to "populations".*

* Done (line 51). Thank you for spotting the typo.

Ln 89. *Here you call it the "pincer" claw but the rest of the manuscript calls it the 'cutter' claw - please be consistent.*

* Done (lines 87-88). We revised the sentence to distinguish between the claw name ("cutter") and the explanation of its function (pincer). Thank you for noticing this inconsistency.

Ln 120. *Remove "very" before "complex".*

* Done (line 132).

Ln 123-126. *It's not clear to me what the difference between the 4th and 5th stage are? Are they only partially benthic in the 4th stage?*

* Exactly. We revised the sentence to clarify this aspect (lines 135-137).

Ln 128. *The start of this sentence is unclear, perhaps it's better to start with "From (month) to (month) 2023, five ovigerous..."*

* Done (line 140).

Ln 128-130. *Were there any differences in the results between locations the mothers were collected from? e.g. did the offspring from the west coast mother have different Ca values?*

* A direct comparison between the groups of offspring from east and west coast origin

would be highly unbalanced in terms of number and variability, so we did not attempt it as it would carry limited meaning since the west coast offspring only come from a single mother. However, we included the identity of the mother in the first full model explaining variation in the coefficient of asymmetry, and found no indication that the origin/ID was relevant in determining the Ca. This was reported in Table 1 in the Appendix/Supplements, with the information related to conditional and marginal R values of the full model. We now added this information explicitly to lines 219-220.

Ln 132. Abbreviation (RAS) should come second after the term and be in parentheses.

* Done (line 144). Thank you for spotting the inconsistency.

Ln 140. Does "they" refer to the mothers?

* Yes, we revised to clarify.

Ln 150. "Grids" haven't been mentioned - do you mean 'tanks'?

* Thank you for spotting this. We referred to the individual holding spaces of the benthic larvae, described in detail in the following section. We revised the sentence to convey this information (lines 162-164).

Ln 153. Missing "the larvae" after "once".

* Thank you, we added it.

Ln 155. Were the four treatments spread across the four different 1500L tanks? e.g. if there was a pathogen present in only one of the tanks could that have affected the outcomes if only one treatment occupied each tank?

* Thank you very much for noticing the ambiguity in the sentence. Indeed, the four treatments were spread equally across the different tanks, for the exact reasons you mention. We added this important specification to lines 168-169 and we added an explicit reference to Fig. 1, where this aspect is clearly illustrated (lines 549-551).

Ln 167-168. The use of "ca" is slightly confusing here because it's the name of your main variable of interest which is also a percentage unit.

* Thank you for noticing, we revised to avoid this confusion (lines 181-182).

Ln 175-177. I had to read this sentence a couple of times to understand - can you please clarify that this means the claw with the larger L/W was the cutter and the claw with the smaller L/W ratio was the crusher.

* Thank you. We revised the sentence using your wording (lines 190-191).

Ln 180-181. I'm being pedantic, but what about exactly 5%?

* Thank you for noticing! We revised the sentence, which now reads "If the asymmetry coefficient was less than 5%, the individual was considered symmetrical, whereas if the asymmetry was equal to or exceeded 5%, the individual was considered asymmetrical (Goldstein and Tlustý, 2003)." (lines 194-195).

Ln 192. Change "model" to "models".

* Done. Thank you for noticing.

Ln 192-193. Was this a Tukey's test? The table caption says Tukey's but here says Bonferroni, can you please clarify. The public R code provided (well done!) suggests it was Tukey's not Bonferroni.

* Thank you for noticing, it was indeed a Tukey's test. The sentence has now been removed in line with the suggestions of Reviewer 2.

Ln 200-201. Were there any differences among groups for the number of excluded lobsters? I.e. did the treatments have an effect on the rate of missing or deformed claws?

* We could not keep a reliable track of this aspect, as missing claws easily fell through the nets to the bottom of the tank or were eaten. However, as we reported in lines 233-235, there was a difference between the groups in terms of mortality, as throughout all the

experimental phase, the group reared with only substrate exhibited lower mortality rates (33.3% of mortality) compared to the other experimental conditions, which instead showed similar values among them (E=46%; SH=45.4%; C=48%).

Ln 202. I assume this is because of logistical/practical reasons, but can you explain why measurements weren't taken at every stage?

* Thank you for noticing this was unclear. Measurements were taken at every stage if individuals were alive, but of course if they died during the time of the experiment we could not get further exuviae. For the individuals that were alive throughout the experiment, measurements were indeed taken at every stage, except in a few cases where the juveniles ate their exuviae before the daily control, a common occurrence. We revised the sentence to clarify (lines 215-217).

Ln 217. Please name the other experimental groups.

*

Done.

Ln 237-241. I think this information is key background knowledge to the study and its underlying assumptions about mechanism of action, so I think it should be moved to the introduction.

* Done (lines 89-93).

Ln 263. Change "claws" to "claw".

*

Done.

Ln 284 - 294. This is a bit repetitive of previous text - it sounds like a conclusion but then more paragraphs follow. I think this could be shortened and moved to the final paragraph or just cut from the manuscript.

* Thank you, we deleted the section.

Ln 304. Make it clear that this is maintenance costs for humans, not the lobsters.

* Done. Thank you for noticing.

Figure 2 - can you add a marker or dashed horizontal line at $y = 5\%$ to indicate the threshold of asymmetry.

* Done.

Table 2- I think this can be moved to supplementary material.

* Done. We moved this information to the Appendix and renamed S1, modified also in line with the comments from Reviewer2.

Reviewer

#2

The authors conducted a randomized controlled trial, whereby a group of juvenile European lobster were randomized to one of 4 experimental conditions (full enrichment comprising the provision of both substrate and shelter; partial enrichment with substrate only; partial enrichment with shelter only and a control condition with no environmental enrichment). The enrichments were introduced at different developmental stages (i.e., the 4th, 5th and 6th benthic stages). The development of claw asymmetry - a critical factor for lobster fitness - was the primary outcome of interest. The results indicated that substrate provision, irrespective of stage of introduction encouraged greater claw asymmetry, and this differentiation was primarily evident from the 6th stage on.

Overall, I enjoyed this article. The biological question is relevant and interesting, the study was well-designed and the interpretation, both in terms of scientific and practical relevance was compelling. Both the design and the statistical analysis are quite complex, and there were some aspects that I believe require further clarification and contextualization. I have made a number

of suggestions below which I hope will be of use to the authors.

* Thank you very much! We are very grateful for your appreciation, and for your constructive comments. We have revised the manuscript accordingly, and we answer in detail to each of your comments below.

Justification of the chosen enrichments:

It would be useful to expand the explanation of the chosen enrichments, and their relevance to both captive and wild- living lobsters. What is the purpose of providing the substrate in the form of oyster shells? Is it because gripping these shells provides the mechanical stimulation necessary to develop claw asymmetry, or do they contain some nutrient that was not available to the juvenile lobsters that were fed the standard diet described on lines 145 - 146?

* Thank you very much! Yes, the choice of oyster shells was based on existing literature showing this effect through mechanical stimulation due to the manipulation of the fragments. We added a more detailed explanation regarding the chosen enrichments (lines 117-120).

I understand that in the wild shelter may be important to protect against predation, but in a controlled environment, what potential benefit does enriching the environment with shelter offer? Perhaps reduced stress levels by providing a place to hide?

* Exactly. The importance of a shelter in captive settings is recognised for key vertebrate models (e.g. Poole 1997; Olsson and Dahlborn 2002; Zhou et al., 2023), but the rationale extends to invertebrates as well. Besides the functional protection from predators, shelters ensure the well-being of animals by allowing the expression of basic innate behaviours, which has direct impact on development and responses to treatments (e.g. Poole 1997; Van der Meeren, 2001; Carere et al., 2014). We added these elements to the explanation above (lines 120-123).

Statistical Analysis:

I found the statistical analysis section somewhat difficult to follow, and suggest that some re-working to enhance clarity may be useful. I write this under the caveat that I am not a statistical expert, and I recognize that this is a complex analysis with many potential applications. As such, I consider these points as suggestions for your consideration rather than as requests.

*Thank you very much for your insights and also for leaving the possibility of considering these as suggestions! We answer each specific point below.

1. Line 191 - 192. *The justification for conducting isolated, in addition to the full model was not clear to me. How did this approach overcome the limitations of the small sample size available in different subsets? Indeed separating points in time could potentially dilute the overall treatment effect of certain conditions, e.g., there is no way that enrichments introduced at stages 5 or 6 could influence timepoint 4 so why test for this? Furthermore, based on the information presented in Table 2, at the 5th stage the treatment effect was non-significant, and at the 4th stage it was pretty close to the pre-defined cut off of 0.05, whereas at the 6th and 7th stage the data are more convincing. This makes sense seeing as theoretically the enrichments would have more time to exert their influence at these later developmental stages. My suggestion would be to work from the full model, and to test specific hypotheses of interest (see below comment 4 about multiple comparisons) based on the results of this model along with the theoretical justification for the study.*

*Thank you for your insights. The isolated analysis is not an addition to the full model, but a follow-up, stemming from the significant interaction between treatment and developmental stage. As such, we would not be able to see how this interaction plays out unless we split the dataset, which we could do either by treatment (and analysing what happened stage by stage), or by stage (allowing us to compare the effects of treatment at every step). We chose the latter, in line with our main research question. So we do work from the full model, but we also need the details expressed in the follow-up analysis to parse out at which point the

treatment starts to be effective, as you also highlighted. We added some more information and details on the rationale of this choice in lines 592-595.

2. I suggest that you include confidence intervals in Figure 3, Panel A. I realise that this will make the figure somewhat messy considering the number of conditions, however in my opinion this graph is the most informative representation of the overall results, but currently the lack of confidence intervals make it difficult to interpret. The dots and error bars could be dodged slightly to allow you to view different conditions that potentially overlap. An alternative suggestion would be to provide the numerical data with confidence intervals in a supplementary file.

*Thank you. We added Panel B to figure 3 exactly to show the patterns of the standard deviations, because adding confidence intervals in Panel A made the figure very messy, to the detriment of the information we wanted to show. We attach below two new figures, which we modified to include the confidence intervals in Panel A. In our opinion, the comparison between the different treatments is less evident in these new figures, which is why we preferred the first version. We hope that after seeing how messy they become, both you and the Editor will agree with us in keeping the original version. However, we are happy with either version of yours and Editor's choice.

3. How were the trend lines for Figure 3, Panel A calculated? Were they estimated from the R emmeans output? If so, the confidence intervals should also be automatically available within the output.

*Trend lines were calculated with the R package `ggplot2`, which was used to generate Figure 3-A. We initially chose not to include the CIs because of concerns regarding the clarity of the resulting figure, as discussed above. However, we included them now, and we are happy with either version you and the Editor prefer.

4. Table 2: Is it really necessary to include so many pairwise comparisons? Considering that you used a post-hoc adjustment, the sheer volume of comparisons likely means that you had very little power to detect potential differences in each one, but some of these comparisons seem unnecessary. For example, there is no need for all comparisons at earlier developmental stages if the enrichments were only offered later. My suggestion would be to theoretically justify the most relevant pairwise comparisons, based on your hypotheses and the outcomes of the full model, rather than to unnecessarily lose power by running all potential comparisons.

A potential complementary analysis could be to conduct an area under the curve analysis, which may overcome some of the limitations of adjusting for multiple comparisons and may provide a

clearer indication of which treatment led to the greatest gain in claw asymmetry. To be clear, I am not suggesting this as a replacement for the main model, but as a potential complement.

***Thank you very much for your insights. We agree, the list of post-hoc comparisons was long and convoluted. We replaced it with the model results from the linear models used to conduct the post-hoc analyses, to investigate the interaction between experimental treatment and developmental stage found in the full model. The results have been moved to the Appendix as Table S1, also in line with the suggestion of Reviewer1.**

General Comments:

Lines 28 - 29 and 48 - 49: The definition provided here seems more appropriate to describe plasticity more generally, whereas I would consider developmental plasticity to be specific to the early-life environment, and how this can shape later-life outcomes.

***Thank you for this attentive comment. We revised the definition to describe specifically developmental plasticity (lines 28-30 and 48-50).**

Lines 407 - 408: These studies by Govind & Pearce, 1989 and Goldstein et al. 2003 appear fundamental to the current one, so it would be useful to more explicitly describe their findings and to explain how the current study builds on these.

***Done. Thank you for suggesting this, we agree it have now improved the contextualisation of our study (Lines 110-113).**

Line 125 - 126: Considering that these developmental stages are core to the study design and interpretation, I suggest that you explicitly define stages 6 and 7 at this point. I assume that these refer to the phases of periodic molting mentioned on line 125? Are the lobsters considered mature at stage 7? It may also be interesting to include the developmental stage in brackets when you later refer to different feeding regimens (Lines 144 - 148). I assume that the planktonic larvae are at stages 1 - 3 and the benthic larvae at stages 4 onward?

***Thank you for your suggestion. We added an explicit reference to stages 6 and 7 in this section (lines 136-137) and added references to developmental stages in relation to feeding regimes (lines 156 and 158). Yes, as we described in lines 135 and 156, the planktonic stages are 1-3, the benthic from 4 onwards as you describe.**

Line 153: Small typo here - should be "once the benthic phase was reached" or "once the juveniles reached the benthic phase".

***Thank you for noticing, we corrected the typo (line 166).**

Line 193: Table 2 indicates that a Tukey, not Bonferonni, adjustment was performed. Which was it?

*** Thank you for noticing, it was a Tukey's test. The sentence has now been removed since pairwise comparisons were replaced with the model results from the linear models used to conduct the post-hoc analyses.**

Line 222: It is not accurate to state that "exposure to physical enrichments significantly influences the development of asymmetric claws" considering that one of the provided enrichments, i.e., shelter, had no influence, and indeed may actually have been counter-productive if it prevented the juvenile lobsters from interacting more with the substrate. I suggest re-phrasing this line to more accurately reflect the main findings.

***Thank you for noticing this oversight. We revised the section accordingly.**

Figure 1: Personally, I believe this Figure is unnecessary as it doesn't seem to add anything beyond what was described in the text. It could even be mis-leading as it implies that there are 12 groups

rather than 10.

***Thank you for your insights. To accommodate possible readers that are more dependent on graphical, schematic representations of experimental designs, we would like to keep the figure. We took care of revising the caption though, to clarify the characteristics (and number) of the experimental groups.**

Second decision letter

MS ID#: bio.061901

MS TITLE: Early-life environment shapes claw bilateral asymmetry in the European lobster (*Homarus gammarus*)

AUTHORS: Lorenzo Latini; Gioia Burini; Valeria Mazza; Giacomo Grignani; Riccardo De Donno; Eleonora Bello; Elena Tricarico; Stefano Malavasi; Giuseppe Nascetti; Daniele Canestrelli; Claudio Carere

I am happy to tell you that your manuscript has been accepted for publication in Biology Open, pending our standard publication integrity checks. It was accepted on 05 Feb 2025.

You are welcome to keep the original version of figure 3.